# Referencing Where to Focus: Improving Visual Grounding with Referential Query

**Yabing Wang** [1], **Zhuotao Tian** [2], **Qingpei Guo** [3],
**Zheng Qin** [1], **Sanping Zhou** [1], **Ming Yang** [3], **Le Wang** [1]*

[1] National Key Laboratory of Human-Machine Hybrid Augmented Intelligence,
National Engineering Research Center for Visual Information and Applications,
and Institute of Artificial Intelligence and Robotics, Xi'an Jiaotong University
[2] Harbin Institute of Technology, Shenzhen, China
[3] AntGroup
{wyb7wyb7,qinzheng}@stu.xjtu.edu.cn
tianzhuotao@link.cuhk.edu.hk
{spzhou, lewang}@xjtu.edu.cn
{qingpei.gqp, m.yang}@antgroup.com

## Abstract

Visual Grounding aims to localize the referring object in an image given a natural language expression. Recent advancements in DETR-based visual grounding methods have attracted considerable attention, as they directly predict the coordinates of the target object without relying on additional efforts, such as pre-generated proposal candidates or pre-defined anchor boxes. However, existing research primarily focuses on designing stronger multi-modal decoder, which typically generates learnable queries by random initialization or by using linguistic embeddings. This vanilla query generation approach inevitably increases the learning difficulty for the model, as it does not involve any target-related information at the beginning of decoding. Furthermore, they only use the deepest image feature during the query learning process, overlooking the importance of features from other levels. To address these issues, we propose a novel approach, called RefFormer. It consists of the query adaption module that can be seamlessly integrated into CLIP and generate the referential query to provide the prior context for decoder, along with a task-specific decoder. By incorporating the referential query into the decoder, we can effectively mitigate the learning difficulty of the decoder, and accurately concentrate on the target object. Additionally, our proposed query adaption module can also act as an adapter, preserving the rich knowledge within CLIP without the need to tune the parameters of the backbone network. Extensive experiments demonstrate the effectiveness and efficiency of our proposed method, outperforming state-of-the-art approaches on five visual grounding benchmarks.

## 1 Introduction

Visual grounding is a challenging multi-modal task that involves localizing a specific object based on a given natural language description. This task requires algorithms to comprehend fine-grained human language expressions and accurately establish correspondences with the target objects. In recent years, it has gained significant attention in research due to its potential for advancing vision-language understanding, such as cross-modal retrieval [42, 30, 44, 9, 41, 43] and image captioning [14, 29].

---

*Corresponding author

38th Conference on Neural Information Processing Systems (NeurIPS 2024).

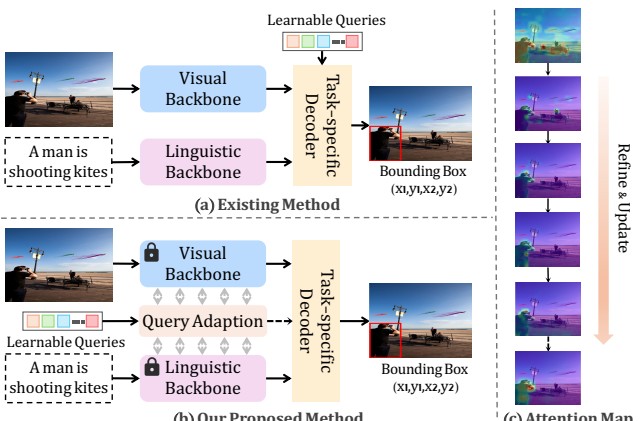

Figure 1: Comparison of DETR-like method and our proposed method for visual grounding. (a) The existing method typically adopts the random initialization queries directly into the decoder to predict the target object. (b) We introduce the query adaption module (QA) to learn target-related context progressively, providing valuable prior knowledge for the decoder. (c) The attention map of the last layer in every QA module and decoder (bottom), respectively.

Existing works in this field typically follow the object detection framework and incorporate multi-modal fusion to tackle this task. Earlier studies [52, 12, 27, 3, 58] mainly focus on a two-stage pipeline, which first generates a set of region proposals using object detectors, and then finds the best-matched region by interacting these regions with linguistic expressions. However, the performance of this method is limited by the quality of the generated region candidates. To address this issue, some studies [48, 47, 4] adopt the one-stage pipeline, which removes the proposal generation stage. Unfortunately, these methods make dense predictions with a sliding window over pre-defined anchor boxes, resulting in sub-optimal performance due to the failure to capture object relations effectively. Recently, some methods [17, 7, 10, 21, 8, 36] inspired by the DETR [1] structure, which adopt a standard multi-modal transformer framework to establish the multi-modal correspondence (as shown in Figure 1 (a)). These methods predict bounding boxes of target objects directly from learnable queries, eliminating the need for extra efforts to obtain candidates, such as region proposals or predefined anchor boxes.

While these methods have shown promising results, their primary focus remains on designing stronger multi-modal decoders. By contrast, much less work has been done to improve the learnable queries, which have been gained extensive attention in the object detection field. The queries that are inputted to the decoder in these methods are typically generated through random initialization or by utilizing linguistic embeddings. We argued that this vanilla approach has two critical issues: **i)** this target-agnostic query inevitably increases the learning difficulty of the decoder. **ii)** During the query learning process, these methods tend to focus solely on the deepest visual features of the backbone, overlooking the texture information that is crucial for the grounding task and present in low and mid-level features, as emphasized by [15, 38].

Drawing from these discussions, this paper seeks to address two critical research questions: **i)** *Can we produce the target-related referential queries for the decoder to alleviate the learning difficulty that the decoder faces?* and **ii)** *How can we effectively incorporate the multi-level visual context information into the query learning process?* We believe that tackling these issues together would promote the learnable query to more comprehensively and accurately learn the corresponding target object information in the image for the visual grounding task.

Considering CLIP [34] carries rich visual-language alignment knowledge, thus we adopt it as the backbone of our approach. Existing methods typically apply CLIP on the visual grounding by fine-tuning its parameters, as CLIP's training objective is to match entire images with text descriptions, rather than capturing fine-grained alignment between regions and textual elements. This may risk losing the general knowledge of CLIP and require significant computational resources. To tackle the challenges mentioned above, we propose a novel approach called RefFormer. Our approach incorporates a query adaptation (QA) module to generate referential queries, which provide the decoder with target-related context (as illustrated in Figure 1 (b)). By strategically inserting QA

module into different layers of CLIP, the query adaptively learns target-related information from multi-level image feature maps, and iteratively refines the acquired information layer by layer. Furthermore, our proposed Reformer can also act as an adapter, enabling CLIP to keep frozen and preserve the original rich knowledge. It adopts the bi-directional interaction scheme, performs the multi-modal fusion by incorporating a small number of trainable parameters, and residually injects new task-specific knowledge into CLIP throughout the entire feature extraction process. Extensive experiments conducted on five popular visual grounding benchmarks (i.e., RefCOCO/+/g [53, 31], Flickr30K [33], and ReferItGame [18]) demonstrate the superior performance of our proposed method.

Our contributions can be summarized as follows: (1) Unlike the previous methods that focus on designing sophisticated multi-modal decoders, we further improve the learning process of the learnable queries, a crucial aspect that has been overlooked in existing work. (2) We propose a query adaption module (QA), which can adaptively capture the target-related context, providing valuable referential knowledge for the decoder. (3) We conduct extensive experiments on five visual grounding benchmarks, demonstrating the effectiveness and potential of our method.

# 2   Related Work

## 2.1   Visual Grounding

Visual grounding aims to ground the target objects based on natural language descriptions by understanding the given images and expressions. Early work [52, 12, 27, 3, 58] primarily focuses on two-stage methods, which formulates the grounding task as a matching task. These methods employ object detectors to generate proposal candidates and then identify the best-matched candidate based on the matching score computed between each proposal and the referring expression. For example, MAttNet [52] proposes to decompose the language expression into three phrase embeddings, which are used to trigger three separate visual modules. While achieving successful performance, two-stage methods heavily rely on the quality of the generated proposals. Based on this, some studies [48, 47, 4] have been dedicated to one-stage methods to remove the proposal generation stage. These methods typically fuse visual features and language features first and then densely regress the bounding box on each position of the feature map grid. For instance, FAOA [48] incorporates linguistic embedding into the YOLOv3 detector to establish a one-stage pipeline, balancing between accuracy and speed.

Recently, transformer-based visual grounding methods [17, 7, 21, 23, 57, 40, 50, 8, 36, 10] have emerged, which leverages the self-attention mechanism to effectively capture intra- and inter-modality relationships and achieve improved performance. Among these methods, the mainstream approach [17, 7, 10, 21, 8, 36] adopts DETR-like structures to decode bounding boxes from learnable queries. For example, Transvg [7] and Transvg++ [8] employ a standard multimodal transformer framework, along with the REG token, to establish multi-modal correspondence and predict the coordinates of the referring object. Notably, the performance improvement of these methods primarily arises from the design of stronger backbones or multi-modal decoders. In this work, we focus on the design of learnable queries, which have received considerable attention in object detection field.

## 2.2   Learnable Queries in DETR and Its Variants

In the object detection field, DETR presents an end-to-end object detection model that is built in an encoder-decoder transformer architecture. However, it suffers from slow training convergence. To address this issue, some follow-up works [55, 19, 49, 45, 20, 26, 24, 54] solve this issue by optimizing the learnable queries in DETR. For instance, Anchor DETR [45] directly treats 2D reference points as queries, while DAB-DETR [24] further investigates the role of queries in DETR and proposes the use of 4D anchor boxes as queries. In contrast to these model-level improvements, DN-DETR [20] introduces query denoising training to mitigate the instability of bipartite graph matching, which is further enhanced by DINO [54].

Additionally, similar research have been explored in other tasks [22, 13, 37]. For example, EaTR [13] formulates a video as a set of event units and treats video-specific event units as dynamic moment queries in video grounding tasks. MTR++ [37] introduces distinct learnable intention queries generated by the k-means clustering algorithm to handle trajectory prediction across different motion modes in motion prediction tasks.

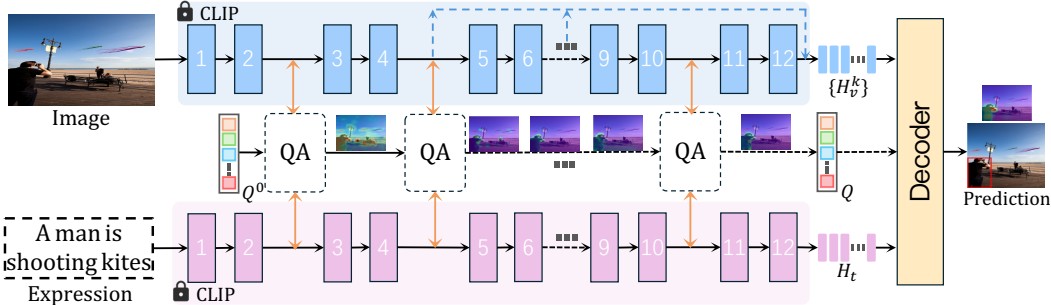

Figure 2: Overview of RefFormer. It adopts a DETR-like structure, consisting of a query adaptation (QA) module that seamlessly integrates into various layers of CLIP, along with a task-specific decoder. By incorporating the QA module, RefFormer can iteratively refine the target-related context and generate referential queries, which provide the decoder with prior context.

## 3   Preliminary

Considering the impressive vision-language alignment capability of CLIP, we take it as the backbone of our method to extract image and text representations, and keep the parameters frozen during training. The feature extraction process can be represented as follows:

**Image Encoder.** For an input image $V \in \mathbb{R}^{H \times W \times 3}$, it is divided into $N$ non-overlapping patches of size $P \times P$, where $N_v = \frac{H \times W}{P^2}$. These patches are then flattened into a set of vectors, represented as $\{\mathbf{x}_v^i \in \mathbb{R}^{3P^2}\}_{i=1}^N$. Next, these vectors are transformed into token embeddings using a linear projection layer $\phi_e(\cdot)$. Furthermore, a classification token $\mathbf{x}_{cls} \in \mathbb{R}^D$ is added at the beginning of the token embeddings. Subsequently, the positional embeddings $\mathbf{E}_v$ are incorporated, and a layer normalization (LN) is applied. This process can be expressed as follows:

$$\mathbf{Z}_v^0 = LN([\mathbf{x}_{cls}; \phi_e(\mathbf{X}_v)] + \mathbf{E}^v) \tag{1}$$

where [;] denotes the concatenate operation. The sequence of tokens $\mathbf{Z}_v^0$ is then passed through $L$ transformer layers. Each transformer layer comprises two submodules: the multi-head self-attention (MHSA) and the multilayer perceptron (MLP), with each submodule preceded by layer normalization.

$$\bar{\mathbf{Z}}_v^i = MHSA(LN(\mathbf{Z}_v^{i-1})) + \mathbf{Z}_v^{i-1}, \ i = 1, ..., L \tag{2}$$
$$\mathbf{Z}_v^i = MLP(LN(\bar{\mathbf{Z}}_v^i)) + \bar{\mathbf{Z}}_v^i \tag{3}$$

where $\mathbf{Z}_v^i \in \mathbb{R}^{N \times D}$ denote the output of $i$-th transformer layer.

**Text Encoder.** Given an referring expression $T$, it is first transformed into a sequence of word embeddings using lower-cased byte pair encoding representations $\mathbf{X}_t$. The word embeddings are bracketed with the [SOS] and [EOS] tokens, producing a sequence of length $N_t$. Similar to the image encoder, these tokens are summed with positional embeddings $\mathbf{E}_t$ and passed through the $L$ transformer layers to extract the text representations:

$$\bar{\mathbf{Z}}_t^i = MHSA(LN(\mathbf{Z}_t^{i-1})) + \mathbf{Z}_t^{i-1}, \ i = 1, ..., L \tag{4}$$
$$\mathbf{Z}_t^i = MLP(LN(\bar{\mathbf{Z}}_t^i)) + \bar{\mathbf{Z}}_t^i \tag{5}$$

where $\mathbf{Z}_t^0 = [\mathbf{x}_{sos}; \mathbf{X}_t; \mathbf{x}_{eos}] + \mathbf{E}_t$, representing the word embedding layer in text encoder.

## 4   Method

The framework is shown in Figure 2. In the following, we first describe our query adaptation module in Section 4.1. We then introduce our decoder that decodes with referential query and training objectives in Section 4.2 and Section 4.3. Furthermore, we extend RefFormer to dense grounding task in Section 4.4. Finally, we provide a discussion in Section 4.5.

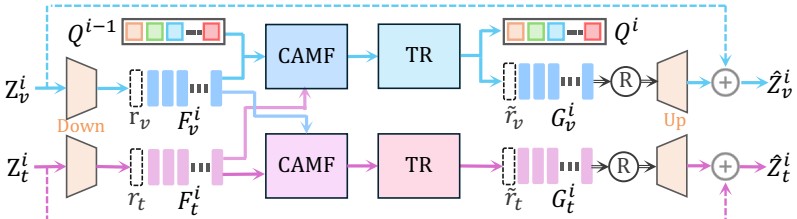

Figure 3: Illustration of our proposed Query Adaption Module, which mainly consists of CAMF and TR modules to generate the referential queries and promote the multi-modal features interaction. "R" represents the feature modulation.

## 4.1 Query Adaptation Module (QA)

In this section, we propose a QA module (as shown in Figure 3) that can generate the referential query to provide the decoder with the target-related context, thereby enhancing the decoder's grounding capabilities. *Importantly, our approach incorporates multi-level features into the query learning process, enabling the queries to capture more comprehensive target object information and can be refined layer by layer.* Furthermore, *QA can also act as an adapter*, eliminating the need to fine-tune the entire parameters of the backbone.

**Down-projection.** Considering the image and language representations $\mathbf{Z}_v^i$ and $\mathbf{Z}_t^i$ obtained from the $i$-th layer of the backbone, we initially use the MLP layers $\phi_{vd}^i(\cdot)$ and $\phi_{td}^i(\cdot)$ to project them to lower-dimensional features to reduce the computation memory:

$$\mathbf{F}_v^i = \phi_{vd}^i(\mathbf{Z}_v^i), \ \mathbf{F}_t^i = \phi_{td}^i(\mathbf{Z}_t^i) \tag{6}$$

**Condition Aggregation and Multi-modal Fusion (CAMF).** We randomly initialize $N_q$ learnable queries $\mathbf{Q} \in \mathbb{R}^{N_q \times D_l}$, where $D_l$ denotes the dimension after projected. These queries are specifically designed to capture potential target object context. Next, we concatenate these queries with the image features and input them, along with the language features into the CAMF block. Specifically, the CAMF block mainly consists of a cross-attention layer, which takes the image and query features $[\mathbf{Q}; \mathbf{F}_v]$ and language features $\mathbf{F}_t$ as the query respectively. This approach enables us to not only incorporate the expression condition into the learnable queries $\mathbf{Q}$ but also to extract relevant information from other modalities, thereby facilitating the fusion of target-related cross-modal features. Besides, we incorporate two learnable regulation tokens $\mathbf{r}_v, \mathbf{r}_t \in \mathbb{R}^{D_l}$ to modulate the final output of each QA. This process can be formalized as follows:

$$\bar{\mathbf{r}}_v, \bar{\mathbf{Q}}_c^i, \bar{\mathbf{F}}_v^i = MHCA([\mathbf{r}_v; \mathbf{Q}^{i-1}; \mathbf{F}_v^i], \mathbf{F}_t^i, \mathbf{F}_t^i) \tag{7}$$

$$\hat{\mathbf{Q}}_c^i = LN(\bar{\mathbf{Q}}_c^i) + \mathbf{Q}^{i-1}, \ \hat{\mathbf{F}}_v^i = LN(\bar{\mathbf{F}}_v^i) + \mathbf{F}_v^i \tag{8}$$

$$\bar{\mathbf{r}}_t, \bar{\mathbf{F}}_t^i = MHCA([\mathbf{r}_t; \mathbf{F}_t^i], \mathbf{F}_v^i, \mathbf{F}_v^i), \ \hat{\mathbf{F}}_t^i = LN(\bar{\mathbf{F}}_t^i) + \mathbf{F}_t^i \tag{9}$$

where $\mathbf{Q}^{i-1}$ represents learnable queries that output from the previous QA, while $\mathbf{Q}^0$ are randomly initialized. The symbol $[;]$ indicates the concatenation operation, and $MHCA(,,)$ and $LN(\cdot)$ denote the multi-head cross-attention layers and layer normalization, respectively.

**Target-related Context Refinement (TR).** Following this, we feed the queries $\hat{\mathbf{Q}}_c$ and multi-modal enhanced feature maps $\hat{\mathbf{F}}_v^i$ and $\hat{\mathbf{F}}_t^i$ into the TR block. First, we use the queries $\hat{\mathbf{Q}}_c$ that have aggregated conditions to interact with the multi-modal enhanced image feature maps $\hat{\mathbf{F}}_v^i$, refining the target-related visual context within them.

$$\mathbf{Q}_v^i = MHCA(\hat{\mathbf{Q}}_c^i, \hat{\mathbf{F}}_v^i, \hat{\mathbf{F}}_v^i), \ \mathbf{Q}^i = LN(MLP(\mathbf{Q}_v^i)) + \hat{\mathbf{Q}}_c^i \tag{10}$$

Moreover, for feature maps $\hat{\mathbf{F}}_v^i$ and $\hat{\mathbf{F}}_t^i$ that have aggregated other modality information, we use the self-attention to further enhance their target-related contextual semantics:

$$\tilde{\mathbf{r}}_v, \widetilde{\mathbf{F}}_v^i = MHSA([\bar{\mathbf{r}}_v; \hat{\mathbf{F}}_v^i], \hat{\mathbf{F}}_v^i, \hat{\mathbf{F}}_v^i), \ \mathbf{G}_v^i = LN(MLP(\widetilde{\mathbf{F}}_v^i)) + \hat{\mathbf{F}}_v^i \tag{11}$$

$$\tilde{\mathbf{r}}_t, \widetilde{\mathbf{F}}_t^i = MHSA([\bar{\mathbf{r}}_v; \hat{\mathbf{F}}_t^i], \hat{\mathbf{F}}_t^i, \hat{\mathbf{F}}_t^i), \ \mathbf{G}_t^i = LN(MLP(\widetilde{\mathbf{F}}_t^i)) + \hat{\mathbf{F}}_t^i \tag{12}$$

**Up-projection.** Finally, we utilize MLP to restore the channel dimension of the image and language features back to their original sizes. These features are then passed as inputs to the next layer of the

backbone in a residual manner. Prior to this, we utilize the regulation token to modulate the features $\mathbf{G}_v$ and $\mathbf{G}_t$, which helps prevent the multi-modal signal from overpowering the original signal.

$$\hat{\mathbf{Z}}_v^i = \phi_{vu}^i(\mathbf{G}_v^i \times \sigma(\widetilde{\mathbf{r}}_v)) + \mathbf{Z}_v^i, \ \hat{\mathbf{Z}}_t^i = \phi_{tu}^i(\mathbf{G}_t^i \times \sigma(\widetilde{\mathbf{r}}_t)) + \mathbf{Z}_t^i \tag{13}$$

where $\phi_{vu}(\cdot)$ and $\phi_{tu}(\cdot)$ denote the MLP layer, and $\sigma(\cdot)$ denotes the sigmoid function.

Finally, by iteratively performing the above process, the queries $Q$ can progressively focus on the target-related context, and generate the referential queries to provide the prior context for the decoder.

## 4.2 Decoding with Referential Query.

**Language-guided Multi-level Fusion.** By inserting the QA at different layers of CLIP, the referential queries can be adaptively updated using the multi-level image feature maps. Additionally, to enhance the image features in decoder, we aggregate the multi-level visual features under the language guidance to yield language-aware multi-level image features. Specifically, given a multi-level image feature set $\{\hat{\mathbf{Z}}_v^k\}$ (including low, mid, and high levels), where $k \in \mathcal{K}$ represents selected layer index, we inject the language features $\mathbf{Z}_t^{last}$ (the final output of the text encoder) into each level of image features using MHCA:

$$\mathbf{H}_{t_{sos}} = \phi_{mt}(\mathbf{Z}_t^{last}), \ \mathbf{H}_v^k = \phi_{mv}^k(\hat{\mathbf{Z}}_v^k) \tag{14}$$

$$\hat{\mathbf{H}}_v^k = MHCA(\mathbf{H}_v^k, \mathbf{H}_{t_{sos}}, \mathbf{H}_{t_{sos}}) + \mathbf{H}_v^k, \ k \in \mathcal{K} \tag{15}$$

where $\phi_{mt}(\cdot)$ and $\phi_{mv}(\cdot)$ denote the linear project function used to map features to the same dimension. Besides, $\mathbf{H}_{t_{sos}}$ represents the [SOS] token in $\mathbf{H}_t$, which extracts the global information of the text. Subsequently, the multi-level language-aware image features are produced by simple concatenation, followed by a linear projection function $\phi_{vml}(\cdot)$ to map to the original dimension:

$$\bar{\mathbf{H}}_{vml} = Concat(\{\hat{\mathbf{H}}_v^k\}), k \in \mathcal{K} \tag{16}$$

$$\mathbf{H}_{vml} = \phi_{vml}(\bar{\mathbf{H}}_{vml}) \tag{17}$$

**Decoding.** Following, we first initialize the queries $\mathbf{Q}'$ with the same size as the referential query $\mathbf{Q}$, and add them together to utilize the prior context in $\mathbf{Q}$. Note that, to avoid interference from $\mathbf{Q}'$ during the initial stage, we initialize $\mathbf{Q}'$ as an all-zero matrix. Then, we concatenate the queries with the image features to interact with the language features to aggregate the condition information and produce the multi-modal feature map $H_{mm}$. This can be represented as:

$$\bar{\mathbf{O}}_c, \bar{\mathbf{H}}_{mm} = MHCA([\phi_q(\mathbf{Q}) + \mathbf{Q}'; \mathbf{H}_{vml}], \mathbf{H}_t, \mathbf{H}_t) \tag{18}$$

$$\mathbf{O}_c = LN(\bar{\mathbf{O}}_c) + \bar{\mathbf{O}}_c, \ \mathbf{H}_{mm} = LN(\bar{\mathbf{H}}_{mm}) + \bar{\mathbf{H}}_{mm} \tag{19}$$

where $\phi_q(\cdot)$ is the MLP layer, which regulates the significance of the query $\mathbf{Q}$. As the importance approaches zero, the query degenerate into a vanilla query. Then, we feed the queries $\mathbf{O_c}$ and multi-modal feature map $\bar{\mathbf{H}}_{mm}$ into the MHCA layer to extract target embddings $\mathbf{O} \in \mathbb{R}^{N_q \times D}$. It can be formulated as:

$$\bar{\mathbf{O}} = MHCA(\mathbf{O}_c, \mathbf{H}_{mm}, \mathbf{H}_{mm}) \tag{20}$$

$$\mathbf{O} = LN(\phi_r(\bar{\mathbf{O}})) + \bar{\mathbf{O}} \tag{21}$$

where $\phi_r(\cdot)$ represents the linear projection function.

**Grounding Head.** We built the two MLPs ($\phi_{box}(\cdot)$ and $\phi_{cls}(\cdot)$) over the target embeddings $\mathbf{O}$. The final outputs consist of the predicted center coordinates of the target object, denoted as $b = (x, y, h, w) \in \mathbb{R}^4$, and the predicted confidence score $y \in \mathbb{R}^2$ that encompass the target object:

$$b = \phi_{box}(\mathbf{O}), \ y = \phi_{cls}(\mathbf{O}) \tag{22}$$

## 4.3 Training Objectives

Similar to DETR, we employ bipartite matching to find the best match between the predictions $\{b, y\}$ and the ground-truth targets $\{b_{tgt}, y_{tgt}\}$. In our case, the class prediction is confidence prediction aims to estimate the confidence of a query containing a target object. To supervise the training, we use the box prediction losses (L1 and GIoU), and a cross-entropy loss after matching.

$$\mathcal{L}_{det} = \lambda_{iou}\mathcal{L}_{iou}(b_{gt}, b) + \lambda_{L1}||b_{gt} - b|| + \lambda_{ce}\mathcal{L}_{ce}(y_{gt}, y) \tag{23}$$

where $\lambda$ denotes the corresponding loss weight. Additionally, to encourage the referential queries in every QA module to effectively focus on the target-related context, we also introduce the auxiliary loss $\mathcal{L}_{aux}$ that is similar to the above objective function to provide supervision for them. The final training objective can be defined as:

$$\mathcal{L}_{final} = \mathcal{L}_{det} + \lambda_{aux}\mathcal{L}_{aux} \qquad (24)$$

where $\lambda_{aux}$ denotes the weight of the auxiliary loss.

### 4.4 Extend to Dense Grounding

In addition to object-level grounding, our method can easily extend to the dense grounding task by incorporating a segmentation head. Specifically, similar to the MaskFormer [5], we utilize the MLP to transform the target embeddings $\mathbf{O}$ into mask embeddings $\mathbf{M} \in \mathbb{R}^{N_q \times D}$. The binary mask prediction $s_i = [0, 1] \in \mathbb{R}^{H \times W}$ is then computed by performing a dot product between the mask embeddings $\mathbf{M}$ and the multi-modal feature map $\mathbf{H}_{mm}$ and followed by a sigmoid activation. During training, we use the mask prediction losses (Focal and Dice), which can be defined as follows:

$$\mathcal{L}_{seg} = \lambda_{focal}\mathcal{L}_{focal}(s_{gt}, s) + \lambda_{dice}\mathcal{L}_{dice}(s_{gt}, s) \qquad (25)$$

where $s_{gt}$ denotes the ground-truth mask.

### 4.5 Discussion

In this work, we aim to explore how to further optimize the learning process of queries. To reduce the learning difficulties posed by vanilla query, we introduce a simple query adaption module to adaptively capture target-related context and iteratively refine it. As illustrated in Figure 5, the attention maps produced by each query adaption module consistently align with our objective: to progressively focus on the target-related context and provide prior context for the decoder. It is worth noting that while "multi-level", "adapter", and "self-attention" may be extensively applied in other research fields, our approach aims to integrate them to address the challenges in visual grounding tasks, instead of designing a specific module to achieve the mentioned functions individually.

## 5 Experiment

### 5.1 Datasets and Evaluation Metric

**RefCOCO/RefCOCO+/RefCOCOg.** RefCOCO [53] comprises 19,994 images featuring 50,000 referred objects, divided into train, val, testA, and testB sets. Similarly, RefCOCO+ [53] contains 19,992 images with 49,856 referred objects and 141,564 referring expressions. It contains more attributes than absolute locations compared to RefCOCO, and has the same split. RefCOCOg [31] has 25,799 images with 49,856 referred objects and expressions. Following a common version of split [32], i.e., train, val, and test sets.

**Flickr30K.** Flickr30k Entities [33] contains 31,783 images and 158k caption sentences with 427k annotated phrases. We follow [7] to split the images into 29,783 for training, 1000 for validation, and 1000 for testing, and report the performance on the test set.

**ReferItGame.** ReferItGame [18] includes 20,000 images with 120,072 referring expressions for 19,987 referred objects. We follow [7] to split the dataset into train, validation and test sets, and report the performance on the test set.

**Evaluation Metric.** For referring expression comprehension (REC), we use Prec@0.5 evaluation protocol to evaluate the accuracy, which is consistent with prior works. In this evaluation, a predicted region is considered correct if its intersection-over-union (IoU) with the ground-truth bounding box is greater than 0.5. For referring expression segmentation (RES), we report the Mean IoU (MIoU) between the predicted segmentation mask and ground truth mask.

### 5.2 Implementation Details

Following [7, 36], the resolution of the input image is resized to $640 \times 640$. We employ the pre-trained CLIP as our backbone to extract both image and language features, and we freeze its parameters

Table 1: Comparisons with the state-of-the-art approaches on three benchmarks, i.e., RefCOCO, RefCOCO+, RefCOCOg. * indicates models that use additionally data beyond RefCOCO series. † indicates that models simply combine RefCOCO, RefCOCO+, and RefCOCOg.

| Methods | Visual Backbone | RefCOCO | | | RefCOCO+ | | | RefCOCOg | |
|---|---|---|---|---|---|---|---|---|---|
| | | val | testA | testB | val | testA | testB | val | test |
| *Single Dataset:* | | | | | | | | | |
| **Two-stage:** | | | | | | | | | |
| MAttNet [52] CVPR2018 | ResNet101 | 76.65 | 81.14 | 69.99 | 69.99 | 71.62 | 56.02 | 66.58 | 67.27 |
| CM-A-E [27] CVPR2019 | ResNet101 | 78.35 | 83.14 | 71.32 | 68.09 | 73.65 | 58.03 | 67.99 | 68.67 |
| Ref-NMS [3] AAAI2021 | ResNet101 | 80.70 | 84.00 | 76.04 | 68.25 | 73.68 | 59.42 | 70.55 | 70.62 |
| PBREC [56] AAAI2024 | ResNet101 | 82.20 | 85.26 | 79.21 | 68.25 | 72.63 | 78.96 | 73.92 | 73.18 |
| **One-stage:** | | | | | | | | | |
| FAOA [48] ICCV2019 | DarkNet53 | 72.54 | 74.53 | 68.50 | 56.81 | 60.23 | 49.60 | 61.33 | 60.36 |
| ReSC-Large [47] ECCV2020 | DarkNet53 | 77.63 | 80.45 | 72.30 | 63.59 | 68.36 | 56.81 | 67.30 | 67.20 |
| MCN [28] CVPR2020 | DarkNet53 | 80.08 | 82.29 | 74.98 | 67.16 | 72.86 | 57.31 | 66.46 | 66.01 |
| PLV-FPN [23] TIP2022 | ResNet101 | 81.93 | 84.99 | 76.25 | 71.20 | 77.40 | 61.08 | 70.45 | 71.08 |
| **Transformer-based:** | | | | | | | | | |
| TransVG [7] ICCV2021 | ResNet101 | 81.02 | 82.72 | 78.35 | 64.82 | 70.70 | 56.94 | 68.67 | 67.73 |
| RefTR [21] NIPS2021 | ResNet101 | 82.23 | 85.59 | 76.57 | 71.58 | 75.96 | 62.16 | 69.41 | 69.40 |
| SeqTR [57] ECCV2022 | DarkNet53 | 81.23 | 85.59 | 76.08 | 68.82 | 75.37 | 58.78 | 71.35 | 71.58 |
| QRNeT [50] CVPR2022 | Swin-small | 84.01 | 85.85 | 82.34 | 72.94 | 76.17 | 63.81 | 71.89 | 73.03 |
| LADS [39] AAAI2023 | ResNet50 | 82.85 | 86.67 | 78.57 | 71.16 | 77.64 | 59.82 | 71.56 | 71.66 |
| TransVG++ [40] TPAMI2023 | ViT-Base/16 | 86.28 | 88.37 | 80.97 | 75.39 | 80.45 | 66.28 | 76.18 | 76.30 |
| Dynamic MDETR [40] TPAMI2023 | ViT-Base/16 | 85.97 | 88.82 | 80.12 | 74.83 | 81.70 | 63.44 | 74.14 | 74.49 |
| VG-LAW [40] CVPR2023 | ViT-Base/16 | 86.06 | 88.56 | 82.87 | 75.74 | 80.32 | 66.69 | 75.31 | 75.95 |
| PVD [6] AAAI2024 | Swin-Base | 84.52 | 86.19 | 76.81 | 73.89 | 78.41 | 64.25 | 74.13 | 71.51 |
| Ours | ViT-Base/32 | 83.97 | 87.80 | 77.45 | 73.55 | 81.09 | 62.24 | 76.33 | 75.33 |
| Ours | ViT-Base/16 | **86.52** | **90.24** | **81.42** | **76.58** | **83.69** | **67.38** | **77.80** | **77.60** |
| *Multiple/Extra Datasets:* | | | | | | | | | |
| VILLA_L * [11] NIPS2020 | ResNet101 | 82.39 | 87.48 | 74.84 | 76.17 | 81.54 | 66.84 | 76.18 | 76.71 |
| RefTR * [21] NIPS2021 | ResNet101 | 85.43 | 87.48 | 79.86 | 76.40 | 81.35 | 66.59 | 78.43 | 77.86 |
| MDETR * [17] ICCV2021 | ResNet101 | 86.75 | 89.58 | 81.41 | 79.52 | 84.09 | 70.62 | 81.64 | 80.89 |
| ShiKra-7B * [2] ARXIV2023 | ViT-Large | 87.01 | 90.61 | 80.24 | 81.60 | 87.36 | 72.12 | 82.27 | 82.19 |
| Ferret-7B * [51] ARXIV2023 | ViT-Large | 87.49 | 91.35 | 82.45 | 80.78 | 87.38 | 73.14 | 83.93 | 84.76 |
| APE † [35] CVPR2024 | ViT-Large | 85.50 | 89.10 | 81.30 | 73.40 | 80.70 | 64.40 | 83.00 | 78.00 |
| Pink * [46] CVPR2024 | ViT-Large | 88.30 | 91.70 | 84.00 | 81.80 | 88.20 | 73.90 | 83.90 | 84.30 |
| Ours † | ViT-Base/16 | 88.82 | 92.52 | 84.87 | 80.91 | 86.64 | 73.35 | 82.29 | 83.15 |
| Ours † | ViT-Large | **90.91** | **93.69** | **86.56** | **83.33** | **89.00** | **75.78** | **84.97** | **84.88** |

Table 2: Comparison with state-of-the-art approaches on the Flickr30K Entities and ReferItGame.

| Methods | Flickr30K | ReferItGame |
|---|---|---|
| | test | test |
| RefTR [21] | 78.66 | 71.42 |
| TransVG [7] | 79.10 | 70.73 |
| QRNet [50] | 81.95 | 74.61 |
| TransVG++ [8] | 81.49 | 74.70 |
| Dynamic MDETR [36] | 81.89 | 70.37 |
| VG-LAW [40] | - | 76.60 |
| Ours | **82.10** | **82.18** |

Table 3: Comparisons with the state-of-the-art dense grounding approaches on three benchmarks for RES task, i.e., RefCOCO, RefCOCO+, and RefCOCOg.

| Methods | RefCOCO | | | RefCOCO+ | | | RefCOCOg | |
|---|---|---|---|---|---|---|---|---|
| | val | testA | testB | val | testA | testB | val | test |
| MAttNet [28] | 56.51 | 62.37 | 51.70 | 46.67 | 52.39 | 40.08 | 47.64 | 48.61 |
| MCN [28] | 62.44 | 64.20 | 59.71 | 50.62 | 54.99 | 44.69 | 49.22 | 49.40 |
| LTS [16] | 65.43 | 67.76 | 63.08 | 54.21 | 58.32 | 48.02 | 54.40 | 54.25 |
| RefTR [21] | 70.56 | 73.49 | 66.57 | 61.08 | 64.59 | 52.73 | 58.73 | 58.51 |
| SeqTR [57] | 67.26 | 69.79 | 64.12 | 54.14 | 58.93 | 48.19 | 55.67 | 55.64 |
| VG-LAW [40] | 75.05 | 77.36 | 71.69 | 66.61 | 70.30 | 58.14 | 65.36 | 65.13 |
| PVD [6] | 74.82 | 77.11 | 69.52 | 63.38 | 68.60 | 56.92 | 63.13 | 63.62 |
| Ours | 74.47 | **77.92** | **72.30** | **66.70** | **72.28** | **60.43** | **65.51** | **66.26** |

during training. The model is optimized end-to-end using AdamW for 40 epochs, with a batch size of 32. We set the learning rate to 1e-4 and the weight decay to 1e-2. The experiments are conducted on V100 GPUs. The loss weight $\lambda_{iou}, \lambda_{L1}, \lambda_{ce}$, and $\lambda_{aux}$, we set to 3.0, 1.0, 1.0, and 0.1. For dense grounding, we set the parameters $\lambda_{focal}$, and $\lambda_{dice}$ to 5.0, and 1.0.

## 5.3 Comparisons with State-of-the-art Methods

**REC Task.** For REC task, we compare the performance with the state-of-the-art REC methods, including the two-stage methods, one-stage methods, and transformer-based methods. As reported in Table 1 and Table 2, our proposed method achieves the best performance. In particular, when comparing to the transformer-based method Dynamic MDETR, which adopts the DETR-like structure and uses the same backbone as ours, we can see that our method performs better with $+0.53\%, +1.90\%, +1.62\%$ on RefCOCO, $+1.11\%, +2.44\%, +6.21\%$ on RefCOCO+, and $+4.94\%, +4.18\%$ on RefCOCOg. Additionally, under multiple/extra datasets setting, our method also surpasses recent state-of-the-art methods that incorporate large language models or utilize more training data.

**RES Task.** Following RefTR [21] and VG-LAW [40], we also conduct the dense grounding experiments and report the results in Table 3 in terms of mIoU. It can be seen that our model

Table 4: Ablation study of the generation method of learnable queries on RefCOCOg.

| Fusion layer ($\mathcal{K}$) | val | test |
|---|---|---|
| {4} | 65.42 (-9.31) | 65.12 (-9.02) |
| {8} | 73.31 (-1.42) | 72.47 (-1.69) |
| {12} | 74.73 (-1.59) | 74.16 (-1.17) |
| {4,8} | 72.71 (-2.02) | 73.03 (-1.13) |
| {4,12} | 75.33 (+0.60) | 74.51 (+0.35) |
| {8,12} | 75.61 (+0.88) | 74.82 (+0.66) |
| {4,8,12} | **76.33 (+1.60)** | **75.33 (+1.17)** |

Table 5: Ablation study of the QA position on RefCOCOg.

| RefFormer layer | val | test |
|---|---|---|
| None | 65.50 | 65.54 |
| {4,8,12} | 74.08 (+8.58) | 73.82 (+8.28) |
| {4,6,8,10,12} | **76.33 (+10.83)** | **75.33 (+9.79)** |
| {2,4,6,8,10,12} | 75.84 (+10.34) | 75.32 (+9.78) |

achieves superior performance without extra deliberate design to dense grounding, demonstrating the generalization of our method.

## 5.4 Ablation Studies

In this section, we conduct the ablation studies to verify the effectiveness the each part of our proposed method on RefCOCOg. Following previous work [7, 17, 8], the visual backbone we apply the ViT-Base/32.

**Effect on the position of QA.** As presented in Table 5, firstly, we can observe that removing the QA would lead to a Sharp decline in performance, highlighting the effectiveness of QA. We then explored the impact of QA's position in the CLIP to determine where QA should be added. We chose three groups for the ablation study: $\{4, 8, 12\}$, $\{4, 6, 8, 10, 12\}$, and $\{2, 4, 6, 8, 10, 12\}$. The results indicate that performance is best when we use the $\{4, 6, 8, 10, 12\}$ configuration. Therefore, we default to this position in our experiments.

**Effect on the layers of multi-level fusion.** In Table 4, we analyze the impact of the fusion layers in the decoder. We first conduct experiments using single-level image features, and then proceed with multi-level features. The results show that utilizing multi-level features significantly improves performance, demonstrating that low- and mid-level features complement high-level features. Additionally, using the $\{4, 8, 12\}$ achieves the best performance, which we adopt for our experiments.

**Effect on the different backbone.** In the first line of Table 6, we apply our method to single-modal encoders, i.e., Swin-Transformer + Bert. The results demonstrate that our method is not only applicable to multi-modal encoders but is also compatible with single-modal encoders.

**Effect on the auxiliary loss.** In the second line of Table 6, We experiment with auxiliary loss, and the results demonstrate the effectiveness of auxiliary loss. By employing auxiliary loss, the reference query can capture the target-related visual contexts more effectively.

**Effect on learnable queries.** In the third line of Table 6, we validate the effectiveness of the learnable queries. Specifically, we replace the prior queries generated by the QA module with randomly initialized queries or linguistic embeddings input to decoder while keeping other modules unchanged. We can observe that introducing the prior queries can bring significant performance improvement. This result demonstrates that prior queries aid the decoder in more accurately locating the target object. Additionally, we investigate the accuracy of our referential queries, which are designed to provide prior information to the decoder. Since the channel dimension in the QA module is lower, the reference query may not accurately predict the coordinates of the targets.

**Convergence curves.** In Figure 4 we illustrate the convergence curve of our proposed method compared to other open-source DETR-like visual grounding methods. Notably, our method demonstrates accelerated training convergence, reducing the training time by half compared to the TransVG, while also outperforming other existing methods.

## 5.5 Qualitative Results

As shown in Figure 5, the attention maps in the QA module illustrate the refined process of how the referential query captures the target-related context. Initially, the attention map appears noisy but gradually focuses on the target-related context, such as the couch in (a). By incorporating the referential query, the attention map in the decoder accurately concentrates on the target object.

Table 6: Ablation studies of backbone, auxiliary loss, and learnable queries on RefCOCOg.

| Method | val | test |
|---|---|---|
| *Backbone:* | | |
| Swin+Bert | 75.25 (-0.64) | **75.61** (+0.29) |
| *Auxiliary loss:* | | |
| W/o $\mathcal{L}_{aux}$ | 74.24 (-1.60) | 73.82 (-1.50) |
| *Learnable queries:* | | |
| Referential query | 52.92 (-22.92) | 51.87 (-23.45) |
| Linguistic embeddings | 71.36 (-4.48) | 71.07 (-4.25) |
| Random initialization | 73.40 (-2.44) | 73.12 (-2.21) |
| Ours | **75.84** | 75.32 |

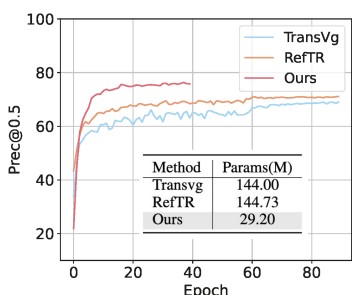

Figure 4: Convergence curves. Our method achieves better results with fewer training epochs on RefCOCOg.

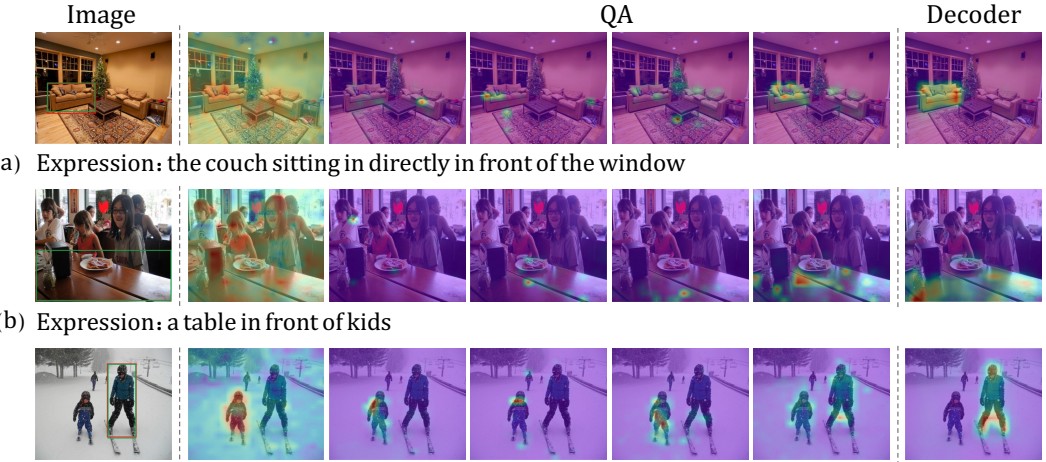

(a) Expression：the couch sitting in directly in front of the window

(b) Expression：a table in front of kids

(c) Expression：a woman skiing with her child

Figure 5: Qualitative results on RefCOCOg. The bounding boxes in green and red correspond to predictions of our model and the ground truth. Columns 2-6 showcase the attention maps generated by each QA module, while the last column represents the attention map from the decoder.

Besides, it is important to note that the referential query may not precisely focus on the target object due to the lower feature dimension in the QA module, but it still captures target-related information.

## 6 Concluding and Remarks

In this paper, we propose a novel approach, called RefFormer that can be seamlessly integrated into CLIP. The RefFormer can not only generate the referential query to provide the target-related context for decoder, but also act as the adaptor to preserve the original knowledge of CLIP and reduce the training cost. Extensive experiments demonstrate the effectiveness of our method, and visualization results illustrate the refined process of our proposed RefFormer.

**Limitations :** Although our method is specifically designed for the REC task and surpasses existing SOTA methods in REC, there is still significant room for improvement in the RES task. This is because we have not yet optimized our approach specifically for the RES task.

## 7 Acknowledgments

This work was supported in part by National Science and Technology Major Project under Grant 2023ZD0121300, National Natural Science Foundation of China under Grants 62088102, 12326608 and 62106192, Natural Science Foundation of Shaanxi Province under Grant 2022JC-41, and Fundamental Research Funds for the Central Universities under Grant XTR042021005.

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

# A  Appendix

Table 7: Ablation study of the direction of the features flow from the QA module on RefCOCOg.

| RefFormer direction | val | test |
|---|---|---|
| None | 65.50 | 65.54 |
| Only text | 70.00 (+4.50) | 69.24 (+3.70) |
| Only image | 72.57 (+7.07) | 72.56 (+7.02) |
| Image & Text | **76.33 (+10.83)** | **75.33 (+9.79)** |

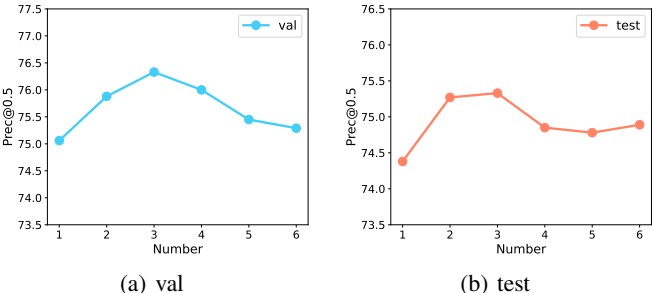

(a) val        (b) test

Figure 6: The performance of different numbers of learnable queries on RefCOCOg.

## A.1  Effect on the RefFormer's direction.

In RefFormer, the QA module can serve as an adapter, injecting specific knowledge into the frozen CLIP model. In Table 7, we investigate the direction of feature flow from QA module. We find that using a dual-direction approach achieves the best performance. Through QA module, language features have aggregated relevant visual context information. As pointed to [25], incorporating rich visual context into linguistic features aids in achieving strong vision-language alignment and better indicating target objects.

## A.2  Effect on the number of learnable queries.

We depict the performance in terms of Prec@0.5 according to the number of learnable queries $N_q$ in Figure 6. When we adopt the $N_q = 3$, the performance is best. However, further increases yield only slight improvements in metrics, as a large number of $N_q$ increases the difficulty of the model. Therefore, we default set the $N_q = 3$ in our experiments.

## A.3  Visualization

Due to space limitations, we present additional visualization results here. As shown in Figure 7, the referential queries gradually focus on the target object and effectively provide target-related context for the decoder. These results demonstrate the effectiveness of our proposed methods.

Image QA Decoder

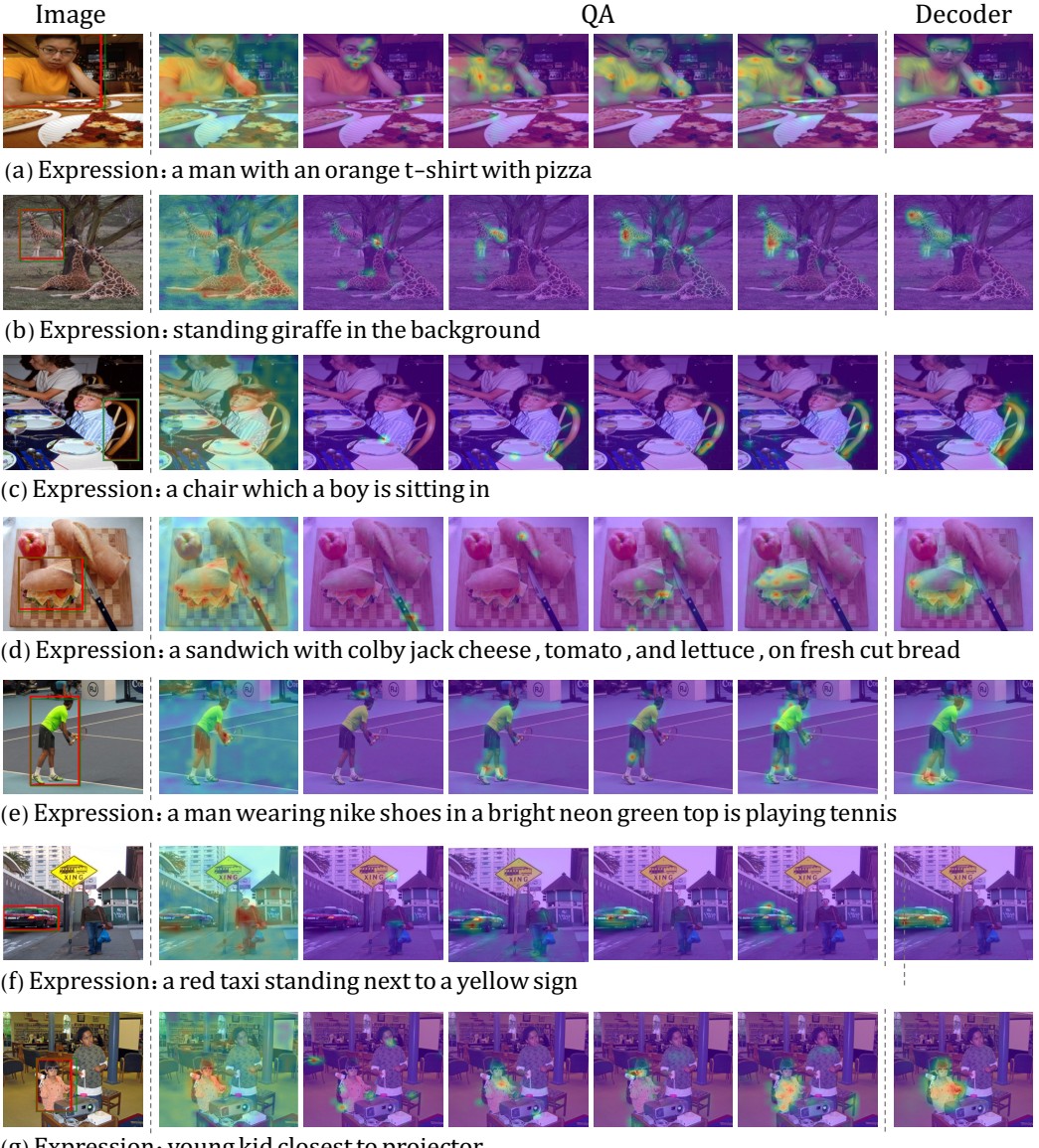

(a) Expression：a man with an orange t-shirt with pizza

(b) Expression：standing giraffe in the background

(c) Expression：a chair which a boy is sitting in

(d) Expression：a sandwich with colby jack cheese, tomato, and lettuce, on fresh cut bread

(e) Expression：a man wearing nike shoes in a bright neon green top is playing tennis

(f) Expression：a red taxi standing next to a yellow sign

(g) Expression：young kid closest to projector

Figure 7: Qualitative results on RefCOCOg. The bounding boxes in green and red correspond to outputs from our model and the ground truth. Columns 2-6 showcase the attention maps generated by each QA module, while the last column represents the attention map from the decoder.

