# OpenReview forum: "Referencing Where to Focus: Improving Visual Grounding with Referential Query"
_NeurIPS.cc/2024/Conference — NeurIPS 2024 poster_

### Official Review · Reviewer_VGeh · 2024-07-07

**Soundness:** 3
**Presentation:** 4
**Contribution:** 4
**Rating:** 7
**Confidence:** 5

**Summary:**

This paper focuses on the visual grounding task, which proposes a query adaption module that can be seamlessly integrated into CLIP.  By strategically inserting this module into different layers of CLIP, the learnable query can adaptively learn target-related information from multi-level image feature maps, and iteratively refine the acquired information layer by layer.  Additionally, it can offer prior information to the decoder, effectively mitigating the learning difficulty of the decoder, and accurately concentrating on the target object.  Extensive experiments on five visual grounding benchmarks (RefCOCO/+/g, ReferItgame, Flickr30k) validate the effectiveness of the proposed method.

**Strengths:**

1.This paper introduces a query adaption module that can adaptively learn target-related information from different layers in CLIP, and provides the visualization results of the attention map in experiments. This design attracted me and made me feel interested.

2.The proposed QA module can not only capture target information adaptively, but also act as an adapter to avoid adjusting Backbone's parameters. Compared with the previous fine-tuning Backbone's work, this module has advantages and achieves better performance with smaller training parameters.

3.In this task, unlike most prior DETR-like research that primarily concentrates on decoder design, this paper focuses on learnable query optimization , making it innovative in its approach.

4.The authors conduct experiments on multiple visual grounding benchmarks (RefCOCO/RefCOCO+/RefCOCOg, Flickr30K, and ReferItGame) and also experiment on the RES benchmark, achieving improved performance. This provides robust empirical support for the effectiveness of the method.

**Weaknesses:**

1.It is better to provide the experiment results without using auxiliary loss, which can further observe the influence of auxiliary loss on the referential query.

2.Is it feasible to directly utilize the referential query without subsequent decoding operations? I am interested in the accuracy of the referential query.

3.Typo: On line 72, "we propose a query adaption module, RefFormer...". It appears that there might be an error in this sentence.

**Questions:**

See weakness, please.  If the author can solve my above confusion well, I will consider raising the score.

**Limitations:**

In the limitations section, the author states that this method still has room for improvement in RES tasks. In addition, by visualizing the attention map, I find that this method accurately locates the area of the target object. Therefore, I believe this method holds great potential in RES tasks, and I hope the author can make further advancements in future work.

---

> ### Author Rebuttal · Authors · 2024-08-06
>
> > Q1:It is better to provide the experiment results without using auxiliary loss, which can further observe the influence of auxiliary loss on the referential query.
>
> We conduct the ablation study with auxiliary loss on RefCOCOg, and the results demonstrate the effectiveness of auxiliary loss. By employing auxiliary loss, the reference query can capture the target-related visual contexts more effectively.
> | Method             | val   | test  |
> |--------------------|-------|-------|
> | w/o auxiliary loss | 74.24 | 73.82 |
> | Ours               | 76.33 | 75.33 |
> |                    |       |       |
>
> Thank you, we will add this experiment to the ablation studies.
>
> > Q2: Is it feasible to directly utilize the referential query without subsequent decoding operations? I am interested in the accuracy of the referential query.
>
>  The referential query is designed to provide prior information to the decoder. Since the channel dimension in the QA module is lower, the reference query may not accurately predict the coordinates of the targets. We conduct the experiment on RefCOCOg, and the results are illustrated below.
>
> | val   | test  |
> |-------|-------|
> | 52.92 | 51.87 |
> |       |       |
>
> Thank you, we will add this explanation to the method.
>
> > Q3: Typo: On line 72, "we propose a query adaption module, RefFormer...". It appears that there might be an error in this sentence.
>
> Thank you. We will rectify "RefFormer" to "QA".

---

> > ### Comment · Reviewer_VGeh · 2024-08-09
> > **Re-response to the author:**
> >
> > Thank the author’s careful response. After carefully reviewing the authors' responses to my concerns and those of other reviewers, the author included a large number of experiments for further explanation and analysis. This has deepened my appreciation for the significance of this paper.
> >
> > The authors have excellently addressed my concerns. As a result, I have decided to raise my score.

---

### Official Review · Reviewer_DDuF · 2024-07-10

**Soundness:** 3
**Presentation:** 3
**Contribution:** 3
**Rating:** 3
**Confidence:** 5

**Summary:**

This paper addresses the generation of queries for the decoder. The authors propose RefFormer to generate the referential query with the prior context. A query adaption module is proposed to capture extensive target-related context and provide valuable referential knowledge for the decoder. Extensive experiments validate the effectiveness of the proposed method.

**Strengths:**

1. This paper proposes a new method for query initialization, where multimodal information is continuously embedded during the feature extraction stage to provide the query with prior knowledge.
2. Extensive experiments validate the proposed method.

**Weaknesses:**

This idea seems straightforward but lacks some innovation. The method mentioned in the paper has been applied in other fields, such as R2-tuning. The proposed module is also simple to implement. Although the model achieves good results, I believe the paper does not meet the standards of NIPS.

**Questions:**

1. Could you provide the number of trainable parameters and the inference speed?

**Limitations:**

Yes. Limitations are addressed.

---

> ### Author Rebuttal · Authors · 2024-08-06
>
> >Q1: This idea seems straightforward but lacks some innovation. The method mentioned in the paper has been applied in other fields, such as R2-tuning.
>
> Thank you for your question. We analyze the differences between our approach and R2-tuning [a] from two aspects: motivation and implementation:
>
> **[Motivation]:** [a] designs an **image-to-video transfer framework** that applies CLIP to video temporal grounding tasks without fine-tuning. In contrast, our work focuses on **improving learnable queries in DETR-like structures** for visual grounding tasks. Specifically, [a] designs a parameter-tuning strategy to model spatial-temporal information from coarse to fine using CLIP, which can be served as an adapter. Instead, our work aims to generate prior information for the decoder to improve the learnable query by integrating the QA module into the CLIP layers.
>
> **[Implementation]:** In our work, we focus on improving learnable queries rather than merely tuning gradient flows. The comparison is shown as follows:
>
> 1. *The feature interaction in CLIP layers.*  It primarily encompasses three key differences: **1) the generated features (frame-level spatial-temporal features vs. referential queries), 2) vision-language interaction (unidirectional vs. bidirectional), and 3) gradient flow tunning (visual side vs. visual and language sides)**. Specifically as follows:
>
>     + [a] introduces the $R^2$ blocks to interact the patch-level representations with language representations for adaptively pool spatial features, and then combine the pooled features with the [CLS] token to extract the frame-level spatial-temporal features. In our work, we incorporate the learnable referential queries to interact with the language and visual features for capturing target-related visual contexts.
>
>    + The interaction in [a] is unidirectional, focusing solely on extracting query (language)-modulated spatial features. Conversely, the interaction in our work is bidirectional, not only highlighting language-related context within visual features but also integrating context-aware visual features into language features.
>
>    + The gradient flows tuning in [a] that forward in CLIP are solely within the vision encoder and are specific to the [CLS] token, while in our method, they are adjusted in both the vision and language encoders and apply to patch features.
>
> 2. *Decoding.*  **[a] apply the traditional convolution layers** to spatial-temporal features to predict the temporal boundary, while **our work adopts the DETR-like structure** and employs **reference queries** to provide the prior information for the decoder.
>
> 3.  *Multi-level features.* We introduce language-guided multi-level fusion to **aggregate the image features from the different layers** rather than applying **1D convolutions to generate the temporal feature pyramid in [a]**  for subsequent decoding.
>
> To further validate the effectiveness of our approach, we integrate the method from [a] into our framework. The results, presented below, demonstrate that our method achieves superior performance.
> | Method                                           | val   | test  |
> |--------------------------------------------------|-------|-------|
> | Decoding with convolution layers                 | 70.02 | 69.87 |
> | Tunning on the visual side and using [CLS] token | 66.97 | 66.12 |
> | Unidirectional interaction                      | 72.45 | 72.02 |
> | Ours                                             | 76.33 | 75.33 |
> |
>
> **Additionally, we would like to highlight that [a] is a concurrent work that has been accepted by ECCV.** We will reference it and discuss it in the related work.
>
> [a] Liu Y, He J, Li W, et al. $ R^ 2$-Tuning: Efficient Image-to-Video Transfer Learning for Video Temporal Grounding[J]. arXiv preprint arXiv:2404.00801, 2024.
>
>
> >Q2: Could you provide the number of trainable parameters and the inference speed?
>
> We provide the number of trainable parameters and the inference speed below and compare them with other open-source methods. When compared to Transvg and RefTR, which also use the DETR-like structure, our approach demonstrates superior performance in both trainable parameters and inference speed.
> | Method  | Trainable parameters(M) | Inference speed(ms) |
> |---------|----------------------|-----------------|
> | ReSC [39]    | 174.85               | 53              |
> | Transvg [7] | 152.61               | 62              |
> | RefTR [19]   | 144.73               | 44              |
> | Ours    | 29.20                | 32              |
> |         |                      |                 |
>
> Thank you, we will add this experiment to the experiments.

---

> > ### Comment · Reviewer_DDuF · 2024-08-13
> >
> > I appreciate the author's rebuttal.  I still think the novelty of this paper is limited. So I keep my score.

---

> ### Author Response · Authors · 2024-08-13
> **Please let us know whether you have any follow-up questions**
>
> Dear Reviewer DDuF,
>
> We hope you are doing well. As the discussion period is coming to an end (Aug 13), we wanted to reach out to see if you have any follow-up questions. If so, we would appreciate the opportunity to respond before the discussion period ends. We believe our above messages should have addressed your concerns, and therefore may warrant an increase in score if you find them helpful as well. Would you please let us know if you have any further questions or concerns? We are happy to continue the discussion.
> Thank you very much again for your thoughtful review and help in improving the paper. We appreciate your time and consideration.
>
> Regards, Authors

---

> ### Author Response · Authors · 2024-08-13
> **Response to Reviewer DDuF**
>
> Thank you for your feedback.  Our approach fundamentally differs from [a]. As previously mentioned, our study concentrates on improving learnable queries within DETR-like architectures, as opposed to solely concentrating on parameter adjustments as in [a]. Furthermore, experiments have demonstrated significant improvements resulting from our approach.
>
> We would like to emphasize once again our motivation:
> 1) We focus on improving the learning process of the learnable queries, different from the previous work that emphasizes the design of sophisticated multi-modal decoders.
> 2) We propose a query adaption module that not only adaptively captures the target-related context, providing valuable referential knowledge for the decoder, but can also serve as the adapter.
> 3) By strategically inserting the QA module into different layers of CLIP, the query adaptively learns target-related information from multi-level image feature maps, and iteratively refines the acquired information layer by layer.
>
> Additionally, we conduct extensive experiments to validate the effectiveness of our method and provide the visualization results to demonstrate the reasonability of our proposed referential query.
>
> We sincerely hope you will reconsider our paper and appreciate your valuable suggestions for improving this work.  If there are any other questions or areas you'd like to discuss, we welcome further conversation.

---

### Official Review · Reviewer_cZhf · 2024-07-10

**Soundness:** 2
**Presentation:** 4
**Contribution:** 2
**Rating:** 6
**Confidence:** 4

**Summary:**

The existing one-stage visual grounding methods suffer from cross-modal learning difficulty and focus simply on the deepest visual features. This paper designs a query adaption (QA) module to provide target-related referential queries for the decoder. The proposed architecture Reformer is based on a CLIP model with multiple QA modules inserted in different layers of CLIP. The proposed method can be applied to both REC and RES tasks. The performance is validated on various benchmarks.

**Strengths:**

-	The paper is generally well-written and easy to follow.
-	The motivation for addressing two issues present in the existing one-stage grounding model is clear, and the idea of enhancing cross-modal interaction is intuitive.
-	The attention maps show the refining process across different layers as expected.
-	It works for both object-level and dense grounding.

**Weaknesses:**

- The proposed module QA has to be inserted into specific positions inside the VLM to improve grounding performance. Experiments in Table 5 indicate the importance of layer selection. However,  it is unclear why the 5 layers perform the best and why these indices are selected. As mentioned in the introduction, the low and mid-level features are crucial for grounding (line 48), yet lower layers (e.g., 2nd) containing rich low-level features diminish the performance. In other words, the selection of layers is not explained theoretically, while 3 combinations are not enough to justify the selection empirically.
- In condition aggregation and multi-modal fusion (CAMF) module, it seems that the query interacts more with the visual representations (i.e., the upper part in Figure 3), while the alternative that mainly integrates query with textual representations seems feasible (i.e., place the query to the lower part in Figure 3). The motivation and advantage of design are missing.
- The backbone and training data information (especially the version of Reformer) is not provided in Tables 2 and 3.

**Questions:**

- How does the paper select the number of inserted QA modules? How does the paper select the indices of inserted layers? Why do more layers or lower layers hurt the performance?
- In the condition aggregation and multi-modal fusion (CAMF) module, the query seems to interact more with the visual representations (i.e., the upper part in Figure 3). Why is it designed in such a way? What is the advantage over mainly based on textual representations?
- Does QA support different model architectures?
- What is the version of the reformer in Tables 2 and 3?

**Limitations:**

While the motivation is clear and the idea is intuitive, the technical implementation of essential modules/structures lacks theoretical or empirical justification.

---

> ### Author Rebuttal · Authors · 2024-08-06
>
> > Q1: How does the paper select the number of inserted QA modules? How does the paper select the indices of inserted layers? Why do more layers or lower layers hurt the performance?
>
>
> We provide more experiments on layer selection on RefCOCOg below. We categorize [1-4], [5-8], and [9-12] as low-level, mid-level, and high-level layers, respectively. As shown in the table below, we observe that introducing more layers can improve performance. However, with continued layer addition, the performance gains become less significant. Therefore, **to strike a balance between performance and computational cost, we opt for [4,6,8,10,12]**. Compared to IDs 7, 9, and 10, line 8 leads to a slight performance decline. This could be attributed to the fact that shallower layers focus on the local details and convey less semantic information, which **may introduce noise**. Similarly, in object detection, RPN-based models [a,b] do not use C1 feature maps for the same reason. Additionally, we explore other combinations of layers, but the performance changes show low sensitivity.
> | ID | Layer          | val   | test  |
> |------|----------------|-------|-------|
> | 1    | None           | 65.50 | 65.54 |
> | 2    | 3,7,11         | 73.94 | 72.94 |
> | 3    | 4,8,12         | 74.08 | 73.82 |
> | 4    | 3,5,7,11       | 74.43 | 74.18 |
> | 5    | 4,6,8,12       | 74.82 | 74.20 |
> | 6    | 3,5,7,9,11     | 75.06 | 74.94 |
> | 7    | 4,6,8,10,12    | 76.33 | 75.33 |
> | 8    | 2,4,6,8,10,12  | 75.84 | 75.32 |
> | 9    | 4,6,8,9,10,12  | 76.40 | 75.31 |
> | 10   | 4,6,8,10,11,12 | 76.51 | 75.43 |
> |      |                |       |       |
>
> Thank you, we will improve Table 5 and the corresponding experimental analysis.
>
> [a] Ren S, He K, Girshick R, et al. Faster R-CNN: Towards real-time object detection with region proposal networks[J]. IEEE transactions on pattern analysis and machine intelligence, 2016, 39(6): 1137-1149.
>
> [b] He K, Gkioxari G, Dollár P, et al. Mask r-cnn[C]//Proceedings of the IEEE international conference on computer vision. 2017: 2961-2969.
>
> > Q2: In condition aggregation and multi-modal fusion (CAMF) module, it seems that the query interacts more with the visual representations (i.e., the upper part in Figure 3), while the alternative that mainly integrates query with textual representations seems feasible (i.e., place the query to the lower part in Figure 3). The motivation and advantage of design are missing.
>
> We would like to clarify that **reference queries aim to capture the target-related visual context.** Initially, they interact with expressions to aggregate text conditions (in CAMF). Subsequently,  they interact with visual features based on text conditions to capture and refine visual contexts about the target object (in TR). If we place them below, we will not be able to capture the target-related visual features based on the conditional information. Additionally, we provide the performance comparison on RefCOCOg as below. The results further demonstrate the effectiveness of the method we designed.
> | Method          | val   | test  |
> |-----------------|-------|-------|
> | Query on the text side| 74.21 | 73.52 |
> | Ours            | 76.33 | 75.33 |
> |                 |       |       |
>
> Thank you, we will add this experiment to the ablation studies.
>
> > Q3: The backbone and training data information (especially the version of Reformer) is not provided in Tables 2 and 3.
>
> For a fair comparison, we utilize the ViT-Base 32 and train on a single dataset (Flickr30K Entities and ReferItGame, respectively) to compare with other methods. Consistent with prior approaches, we perform ablation studies (Table 3-6) using ViT-Base 32 and train on a single dataset (RefCOCOg).
>
> Thank you, we will further improve the experimental details in our paper.
>
> > Q4: Does QA support different model architectures?
>
> We apply our method to single-modal encoders, i.e., Swin-base + Bert, and conduct experiments on RefCOCOg. The results, as presented below, demonstrate that our method is also compatible with single-modal encoders.
>
> | Method          | val   | test  |
> |-----------------|-------|-------|
> | Swin-base + BERT| 75.25 | 75.61 |
> | CLIP-base           | 76.33 | 75.33 |
> |                 |       |       |
>
> Thank you, we will add this experiment to the experiments.

---

> > ### Comment · Reviewer_cZhf · 2024-08-12
> >
> > The authors have addressed my concerns, and it would be better if they could include the supplement experiments in their revision. I tend to accept the paper after the rebuttal.

---

### Official Review · Reviewer_RwV9 · 2024-07-11

**Soundness:** 4
**Presentation:** 3
**Contribution:** 4
**Rating:** 7
**Confidence:** 5

**Summary:**

This paper proposes a novel visual grounding framework, called RefFormer, aims to improve the learning process of learnable queries. Specifically, it introduces a  query adaption module (QA) that can be seamlessly integrated into different layers of CLIP, which can not only provide prior information to the decoder, but also act as an adapter to learn task-specific knowledge. The effectiveness of the proposed method is extensively validated through experiments conducted on the five popular visual grounding benchmarks, namely RefCOCO, RefCOCO+, RefCOCOg, Flickr30K, and ReferItGame. Furthermore, the authors extend this approach to dense grounding tasks, demonstrating its effectiveness and generalization.

**Strengths:**

1. The proposed method is novel, which integrates query learning and adapter into one module, effectively leveraging various levels of feature information within CLIP.
2. By introducing QA module, models can adaptively extract target-related information from the backbone and provide prior information to the decoder instead of relying on randomly initialized queries.
3. This paper achieves better performance with fewer training parameters and data.
4. Sufficient experiments have been conducted to show the promising results of the proposed method.
5. The writing of this paper is good and clear.

**Weaknesses:**

While the paper is clearly written, some areas can still be improved. Some suggestions are as follows：
1. Line 72 should correct "RefFormer" to "QA".
2. A more detailed description is needed for the text side in the QA module, such as how the interaction with visual features is initiated.
3. Why is a direct use of the referential query not preferred, and why is an additional learnable query introduced in the decoder?
4. Specify the dimension along which the concatenation operation in Section 4.1 is performed.
5. Correct "[r_t, F^i_t]" to "[r_t; F^i_t]" in Equation 9.
6. Provide experimental details regarding the channel dimension of the QA module in the down-projection process.

**Questions:**

1. Why is a direct use of the referential query not preferred, and why is an additional learnable query introduced in the decoder?
2. 6. Provide experimental details regarding the channel dimension of the QA module in the down-projection process.

**Limitations:**

See the above weaknesses.

---

> ### Author Rebuttal · Authors · 2024-08-06
>
> > Q1: Line 72 should correct "RefFormer" to "QA".
>
>  Thank you.  We will correct the typos in our paper.
>
> > Q2: A more detailed description is needed for the text side in the QA module, such as how the interaction with visual features is initiated.
>
>  In the CAMF block, we take the language features as the query, and interact them with the image features using cross-attention. By doing so,  we can incorporate rich visual context into language features to better indicate the target object.  Subsequently, in the TR block, we employ self-attention to enhance the context-aware language features produced above to refine the expression condition.
>
> Thank you, we will add this description in Sec 4.1.
>
> > Q3: Why is a direct use of the referential query not preferred, and why is an additional learnable query introduced in the decoder?
>
> As outlined in line 189, we feed the referential query into $\phi_q(\cdot)$ to adjust its significance. When the referential query is inaccurate (i.e., its significance approaches zero), the query in the decoder degenerates to the vanilla query.
>
> Thank you, we will add this explanation in line 189.
>
> > Q4: Specify the dimension along which the concatenation operation in Section 4.1 is performed.
>
> Thank you for your suggestion. We will further improve the description details in our paper. In Eq.7,9,11,12, the concatenation is along with the patch dimension.
>
> > Q5: Correct "[r_t, F^i_t]" to "[r_t; F^i_t]" in Equation 9.
>
>  Thank you.  We will correct the typos in our paper.
>
> > Q6: Provide experimental details regarding the channel dimension of the QA module in the down-projection process.
>
>  Thank you for your suggestion. In our experiments, we set the channel dimension of the down-projection to 128. Furthermore, we provide an ablation study on the channel dimension of the down-projection below. Increasing the dimension improves performance, but to strike a balance between performance and computational cost, we set the channel dimension to 128.
>
> | Dimension | val   | test  |
> |-----------|-------|-------|
> | 64        | 74.15 | 73.88 |
> | 128       | 76.33 | 75.33 |
> | 256       | 76.62 | 75.39 |
> |           |       |       |
>
> Thank you, we will add this experiment to the ablation studies.

---

> ### Author Response · Authors · 2024-08-13
> **Please let us know whether you have any follow-up questions**
>
> Dear Reviewer RwV9,
>
> I am glad for the recognition of our work. Please feel free to raise any further questions, and we are more than happy to continue the discussion with you. Thanks again for your great efforts and constructive advice in reviewing this paper!
>
> Regards, Authors

---

> > ### Comment · Reviewer_RwV9 · 2024-08-13
> > **Comment to the author**
> >
> > Thanks to the author's response, which has effectively addressed my concerns regarding the details of this paper. Overall, I find this work to be impressive, I am inclined to maintain my recommendation to accept it. I hope the author can address the aforementioned issues, as well as those raised by other reviewers, in the revised version.

---

### Decision · Program_Chairs · 2024-09-25

**Decision:**

Accept (poster)

**Comment:**

The paper received mixed feedback, with one reviewer recommending rejection while three others favored acceptance. The authors proposed a novel approach called RefFormer, introducing a query adaptation module designed to learn target-related context more effectively. Although Reviewer DDuF pointed out that the idea appears straightforward and lacks significant innovation, noting similarities to existing methods. During rebuttal, the authors provided detailed comparison between their paper and the mentioned ECCV 24 work, highlighting differences in motivation and implementation. Additionally, they integrated the mentioned method into their framework for experimental analysis. The AC believed the rebuttal is persuasive and clearly demonstrate the novelty and innovation of the proposed method. Therefore, the AC has decided to accept this paper.